# Changes in mRNA abundance drive shuttling of RNA binding proteins, linking cytoplasmic RNA degradation to transcription

Sarah Gilbertson[1], Joel D Federspiel[2], Ella Hartenian[1], Ileana M Cristea[2], Britt Glaunsinger[1,3,4]*

[1]Department of Molecular and Cell Biology, University of California, Berkeley, United States; [2]Department of Molecular Biology, Princeton University, Princeton, United States; [3]Department of Plant & Microbial Biology, University of California, Berkeley, United States; [4]Howard Hughes Medical Institute, United States

**Abstract** Alterations in global mRNA decay broadly impact multiple stages of gene expression, although signals that connect these processes are incompletely defined. Here, we used tandem mass tag labeling coupled with mass spectrometry to reveal that changing the mRNA decay landscape, as frequently occurs during viral infection, results in subcellular redistribution of RNA binding proteins (RBPs) in human cells. Accelerating Xrn1-dependent mRNA decay through expression of a gammaherpesviral endonuclease drove nuclear translocation of many RBPs, including poly(A) tail-associated proteins. Conversely, cells lacking Xrn1 exhibited changes in the localization or abundance of numerous factors linked to mRNA turnover. Using these data, we uncovered a new role for relocalized cytoplasmic poly(A) binding protein in repressing recruitment of TATA binding protein and RNA polymerase II to promoters. Collectively, our results show that changes in cytoplasmic mRNA decay can directly impact protein localization, providing a mechanism to connect seemingly distal stages of gene expression.
DOI: https://doi.org/10.7554/eLife.37663.001

*For correspondence:
glaunsinger@berkeley.edu

Competing interests: The authors declare that no competing interests exist.

## Introduction

mRNA decay is a critical stage of gene expression that regulates the abundance and lifespan of cellular mRNAs. Many viruses including alpha and gammaherpesviruses, influenza A virus, and SARS coronavirus accelerate host mRNA degradation through the use of viral proteins that trigger endonucleolytic cleavage of mRNAs in the cytoplasm. Each of these viral proteins bypasses the rate-limiting deadenylation step of the basal decay pathway, resulting in cleaved mRNAs that are rapidly degraded by the major cellular 5′−3′ exonuclease Xrn1 (*Covarrubias et al., 2011*; *Gaglia et al., 2012*). This process, termed 'host shutoff', allows viruses to rapidly restrict cellular gene expression in order to blunt immune responses and liberate resources for viral replication (*Abernathy and Glaunsinger, 2015*; *Burgess and Mohr, 2015*; *Gaglia and Glaunsinger, 2010*; *Rivas et al., 2016*). Viral endonucleases have also served as tools for deciphering how cells sense and respond to large changes in mRNA abundance.

While mRNA decay is often considered the terminal stage of gene expression, the rate of mRNA decay has recently been shown to influence transcription by RNA polymerase II (RNAPII) in both yeast and mammalian cells (*Abernathy et al., 2015*; *Braun and Young, 2014*; *Haimovich et al., 2013*; *Sun et al., 2013*). In yeast, a buffering system exists in which Xrn1 plays a major role in connecting mRNA synthesis and decay, presumably allowing cells to maintain an appropriate overall

**eLife digest** The nucleus of a cell harbors DNA, which contains all information needed to build an organism. The instructions are stored as a genetic code that serves as a blueprint for making proteins – molecules that are important for almost every process in the body – and to assemble cells. But first, the code on the DNA needs to be translated with the help of a 'middle man', known as messenger RNA. These molecules carry information to other parts of the cell, wherever it is needed.

Messenger RNA is produced in the nucleus of a cell, and then exported into the material within a cell, called the cytoplasm, as a template to produce proteins. Once this process has finished, the template is destroyed. The rate at which the messenger RNA is made affects the flow of genetic information.

However, recent evidence suggests that the speed at which messenger RNA is destroyed in the cytoplasm can influence how much of it is made in the nucleus, i.e., if high levels of RNA are destroyed, the production is stopped. For example, it has been shown that certain viruses possess proteins that speed up the destruction of messenger RNA to gain control over the host cell.

Here, Gilbertson et al. wanted to find out more about how the breakdown of RNA can signal the nucleus to stop producing these molecules. Messenger RNAs are coated with proteins, which are released when the RNA is destroyed. To test if some of those proteins travel back to the nucleus to influence the production of messenger RNA, proteins in human cells grown in the laboratory were labeled with specific trackers. RNA destruction was induced, in a way that is similar to what happens during a virus attack. The experiments revealed that many RNA-binding proteins indeed return to the nucleus when RNA is destroyed. One of these proteins, named cytoplasmic poly(A)-binding protein, played a key role in transmitting the signal between the cytoplasm and the nucleus to control the production messenger RNA.

The amount of messenger RNA can change in many ways throughout the life of a cell. For example, viral infections can lower it and limit the growth and health of cells. A drop in these molecules could act as an early warning of ill health in cells and trigger responses in the nucleus. This new link between messenger RNA destruction and production may help to shed new light on how cells use different signals to control the production of their own genes while restricting pathogens from taking over. A next step will be to determine how these signals communicate with the RNA production machinery in the nucleus and how certain viruses can subvert this process to activate their own genes.

DOI: https://doi.org/10.7554/eLife.37663.002

mRNA abundance. Mammalian cells also have a mechanism to sense mRNA levels, though the pathway appears to operate differently than in yeast. Here, accelerated cytoplasmic mRNA degradation does not lead to a compensatory increase in mRNA synthesis, as might be predicted by the homeostatic model, but instead decreases cellular RNAPII promoter recruitment, thereby amplifying the restrictive gene expression environment (*Abernathy et al., 2015*). Significant transcriptional repression as measured by nascent mRNA production was reported to occur at approximately 9% of host genes, although validation experiments suggested this number is likely to be an underestimate (*Abernathy et al., 2015*).

The mRNA decay-transcription feedback pathway is activated in mammalian cells infected with gammaherpesviruses like Kaposi's sarcoma-associated herpesvirus (KSHV) and murine gammaherpesvirus 68 (MHV68), as well as upon expression of virally encoded mRNA endonucleases in uninfected cells. Herpesviral endonucleases, including the muSOX protein of MHV68, cleave mRNA but do not impact the abundance of noncoding RNAs transcribed by RNA polymerase I (RNAPI) or III (RNAPIII) (*Covarrubias et al., 2011*). Correspondingly, muSOX-induced mRNA decay elicits a significant decrease in RNA polymerase II (RNAPII) recruitment to cellular promoters (*Abernathy et al., 2015*). Notably, depletion of Xrn1 from muSOX-expressing cells prevents the ensuing RNAPII transcriptional repression. This suggests that the initial mRNA cleavage and translational inactivation are insufficient to restrict RNAPII recruitment, and that subsequent exonucleolytic degradation of the cleaved mRNA fragments is a critical signaling step.

Little is currently known about this pathway linking cytoplasmic mRNA decay to RNAPII activity in mammalian cells, including the nature of the signal that is transmitted between the two cellular compartments. An attractive hypothesis is that one or more RNA binding proteins (RBPs) differentially traffics between the cytoplasm and the nucleus when basal rates of mRNA decay are perturbed, thereby conveying global mRNA abundance information. Recent analyses indicate that mammalian cells contain hundreds of RBPs that bind polyadenylated mature mRNAs, and proteins within this group have been shown to regulate all stages of gene expression (*Gerstberger et al., 2014*; *Mitchell and Parker, 2014*; *Müller-McNicoll and Neugebauer, 2013*; *Singh et al., 2015*). Furthermore, RBPs frequently display nucleocytoplasmic shuttling behavior.

Here, we charted global alterations in protein localization that occur specifically in response to increased or decreased Xrn1 activity. This revealed a set of mammalian RBPs that preferentially move from the cytoplasm to the nucleus during accelerated mRNA decay, as well as components of the 5′−3′ decay machinery and other RBPs whose subcellular distribution is altered in cells lacking Xrn1. Poly(A) tail associated proteins are overrepresented among the RBPs that accumulate in the nucleus under conditions of global mRNA decay, offering an explanation for how RNAPII could be selectively sensitive to mRNA abundance. Indeed, we uncovered a new role for cytoplasmic poly(A) binding protein (PABPC) in mediating mRNA decay-driven repression of RNAPII promoter recruitment. Furthermore, we show that the recruitment of TATA binding protein (TBP) to promoters is also impaired in response to PABPC nuclear translocation, indicating that cytoplasmic mRNA decay impacts early events in preinitiation complex assembly. Our results reveal how mRNA levels exert significant influence on RBP localization and suggest that select RBPs transmit mRNA abundance information from the cytoplasm to the nucleus to broadly influence gene expression, particularly under conditions of cellular stress.

## Results

### RNA binding proteins translocate from the cytoplasm to the nucleus in cells undergoing enhanced cytoplasmic mRNA decay

To chart mRNA decay-driven movement of proteins between the cytoplasm and the nucleus, we used a quantitative liquid chromatography/tandem mass spectrometry (LC/MS-MS)-based approach. Specifically, following subcellular fractionation, proteins from nuclear and cytoplasmic fractions were labeled with isobaric tandem mass tags (TMT). TMT labeling enables multiplexing of up to 11 samples per run and was proven to improve the analytical power for quantitation during viral infections (*McAlister et al., 2012*; *Jean Beltran et al., 2016*). We used HEK293T cells expressing the MHV68 muSOX endonuclease to create a condition of accelerated, Xrn1-dependent cytoplasmic mRNA decay. We previously demonstrated that muSOX expression in these cells activates the mRNA decay-RNAPII transcription feedback pathway similar to virally infected fibroblasts (*Abernathy et al., 2015*). Pure populations of cells expressing either WT muSOX or the catalytically dead D219A muSOX point mutant were generated using Thy1.1-based cell sorting. Here, muSOX was fused to the cell surface glycoprotein Thy1.1 with an intervening self-cleaving 2A protease, causing release of Thy1.1 from muSOX for cell surface expression and selection. Three biological replicates of control, WT, and D219A muSOX expressing cells were then separated into nuclear and cytoplasmic fractions, and trypsin-digested proteins from each fraction were differentially TMT labeled prior to LC/MS-MS (*Figure 1A*).

Among the 5994 total quantifiable nuclear proteins (detected in all replicates), 123 displayed significant nuclear enrichment (adjusted P value of $\leq 0.05$) in WT muSOX expressing cells relative to the D219A mutant (*Figure 1B*, Table S1-2 in *Supplementary file 1*). We then removed from further analysis proteins that were simultaneously increased in the cytoplasm in muSOX expressing cells to remove proteins that increase in overall abundance, as well as proteins displaying significant differences between the D219A catalytic mutant and the empty vector control. These filtering steps yielded a final list of 67 proteins that were differentially enriched in the nucleus under conditions of accelerated mRNA decay (*Figure 1B*, Table S3 in *Supplementary file 1*). Notably, 22 of the 67 proteins (33%) are annotated as RBPs (Pantherdb) in line with the expectation that mRNA-bound proteins in particular should be impacted during widespread mRNA degradation. In addition, 31 of the 67 proteins (46%) are listed as localized both to the cytoplasm and nucleus according to the

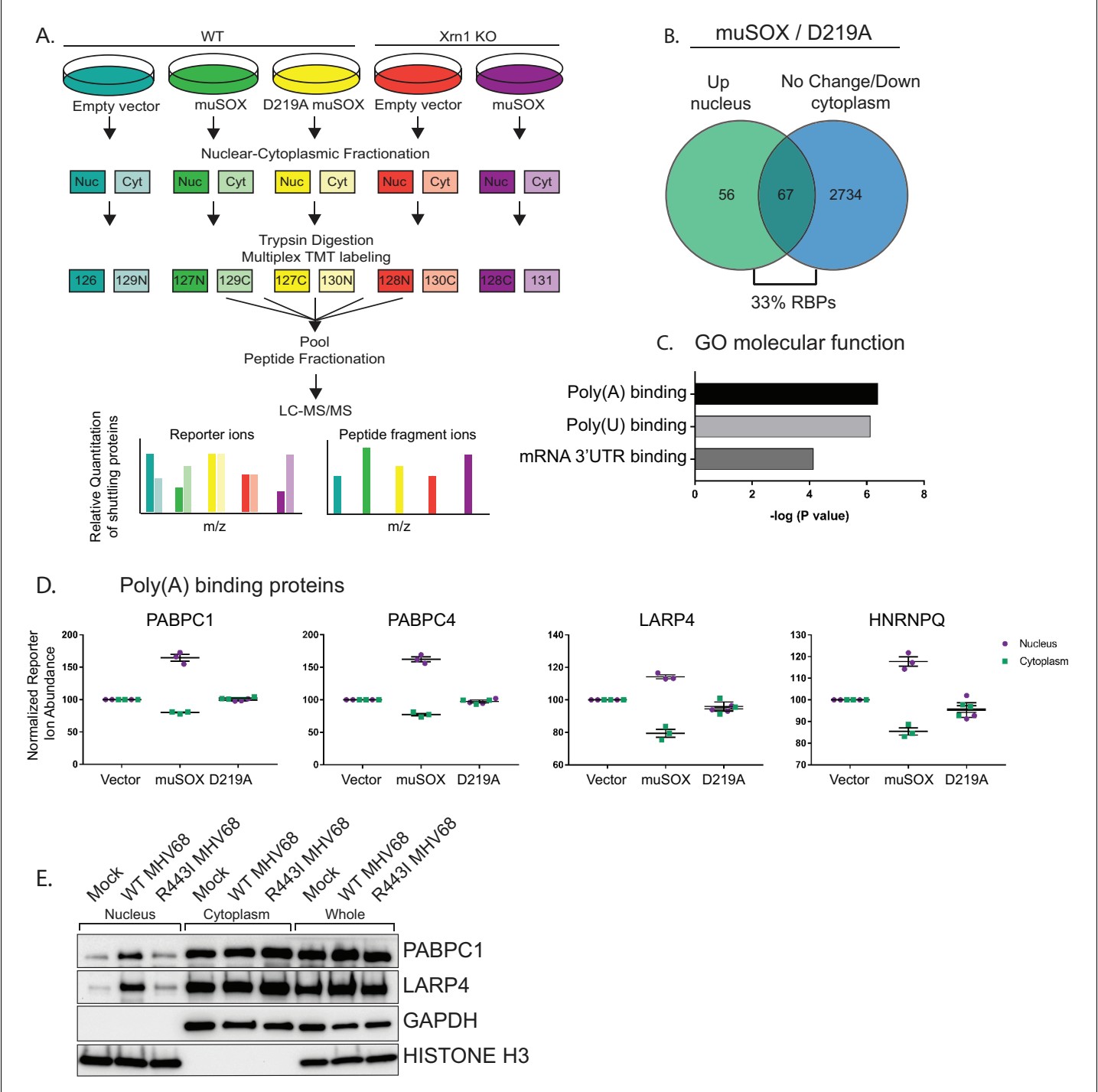

**Figure 1.** RNA binding proteins are translocated from the cytoplasm to the nucleus in cells undergoing enhanced cytoplasmic mRNA decay. (**A**) Diagram depicting the experimental setup. (**B**) Venn diagram of nuclear proteins that are specifically and significantly (p<0.05) enriched in muSOX-expressing cells compared to D219A-expressing cells that also show either no change or a decrease in cytoplasmic abundance. (**C**) Gene ontology molecular function overrepresentation analysis by Pantherdb, graphed according to their P value. (**D**) Graphs showing the nuclear and cytoplasmic distribution of poly(A) binding proteins from the TMT-MS data. Graphs display the mean with SEM of 3 biological replicates. (**E**) Western blot of nuclear, cytoplasmic, and whole cell fractions of NIH3T3 fibroblasts mock infected or infected with WT or R443I MHV68 for 24 hr. GAPDH and histone H3 serve as fractionation and loading controls. Shown is a representative example of 3 biological replicates.

DOI: https://doi.org/10.7554/eLife.37663.003

The following figure supplement is available for figure 1:

*Figure 1 continued on next page*

*Figure 1 continued*

**Figure supplement 1.** RNA binding proteins are translocated from the cytoplasm to the nucleus in cells undergoing enhanced mRNA decay.

DOI: https://doi.org/10.7554/eLife.37663.004

Database for Annotation, Visualization and Integrated Discovery (DAVID), supporting the idea that they are shuttling factors. As an independent validation of these results, we evaluated 12 of the top hits by western blotting of fractionated cell lysates in control or muSOX-expressing cells, 10 of which recapitulated the MS results (*Figure 1—figure supplement 1A*).

## Proteins associated with the poly(A) tail display robust mRNA decay-dependent nuclear translocation

The poly(A) tail is a defining mRNA feature and during basal mRNA decay, deadenylation is the initiating step that licenses subsequent decapping and exonucleolytic degradation of an mRNA (*Schoenberg and Maquat, 2012*). Thus, the binding state of poly(A) tail associated proteins could theoretically serve as a readout to distinguish the overall abundance of mRNA over other forms of RNA in the cytoplasm. Notably, nuclear relocalization of PABPC has been observed during infection with multiple viruses that promote mRNA decay, supporting the idea that poly(A) tail associated proteins may be particularly sensitive to mRNA abundance (*Harb et al., 2008*; *Lee and Glaunsinger, 2009*; *Park et al., 2014*; *Piron et al., 1998*; *Salaun et al., 2010*). Indeed, an overrepresentation analysis using Pantherdb revealed that poly(A) binding proteins, poly(U) binding proteins, and mRNA 3'UTR binding proteins were significantly overrepresented among the 67 differentially expressed proteins (*Figure 1C*, *Figure 1—figure supplement 1B–C*). Proteins linked to the poly(A) tail consistently arose as robust hits in our MS dataset, including PABPC proteins 1 and 4 (PABPC1, PABPC4), LA-related protein 4 (LARP4), and heterogeneous nuclear ribonucleoprotein Q (HNRNPQ) (Table S3 in *Supplementary file 1*, *Figure 1D*). We confirmed that PABPC1 and LARP4 also translocate to the nucleus in NIH3T3 cells infected with WT MHV68, but not in cells infected with an MHV68 muSOX mutant virus (R443I) with impaired mRNA cleavage activity (*Adler et al., 2000*; *Richner et al., 2011*) (*Figure 1E*). Thus, poly(A) associated proteins preferentially move from the cytoplasm to the nucleus in response to muSOX-activated mRNA decay in both transiently transfected and virally infected cells.

## Nuclear translocation of RNA binding proteins is dependent on mRNA degradation by Xrn1

Xrn1 is the major 5'−3' exonuclease in mammalian cells and is responsible for the degradation of 3' RNA fragments generated upon cleavage by muSOX (*Gaglia et al., 2012*). In the absence of Xrn1, muSOX-induced repression of RNAPII promoter occupancy does not occur, suggesting that Xrn1 activity should be required for release and subsequent nuclear translocation of RBPs involved in this phenotype (*Abernathy et al., 2015*). We therefore used Cas9-based genome editing to generate Xrn1 knockout clones in HEK293T cells and confirmed that muSOX expression in these cells failed to reduce RNAPII promoter occupancy (*Figure 2—figure supplement 1A–B*). The Xrn1 knockout cells exhibited a ∼ 2 fold reduction in growth compared to control Cas9-expressing WT cells (*Figure 2—figure supplement 1C*), in line with observations in yeast (*Larimer and Stevens, 1990*). Importantly, this did not lead to broad changes in gene expression (see below). Given that Xrn1 is a central component of the mammalian mRNA decay machinery, only low passage versions of these cells were used to decrease the likelihood of compensatory changes occurring in other decay components. Using the same TMT-LC/MS-MS strategy described above, we analyzed nuclear and cytoplasmic fractions from three biological replicates of Xrn1 knockout cells expressing muSOX or an empty vector control (*Figure 1A*). Comparison of these data to the list of proteins from Table S3 in *Supplementary file 1* indicated that 45 of the 67 hits failed to shuttle in muSOX-expressing Xrn1 knockout cells, confirming that our workflow identified factors that differentially shuttle in response to mRNA degradation (*Figure 2A,B*).

Poly(A) tail degradation is normally carried out by deadenylases prior to activation of Xrn1-mediated decay from the 5' end, but we previously demonstrated that SOX-cleaved mRNAs are not deadenylated prior to their targeting by Xrn1 (*Covarrubias et al., 2011*). Indeed, analysis of

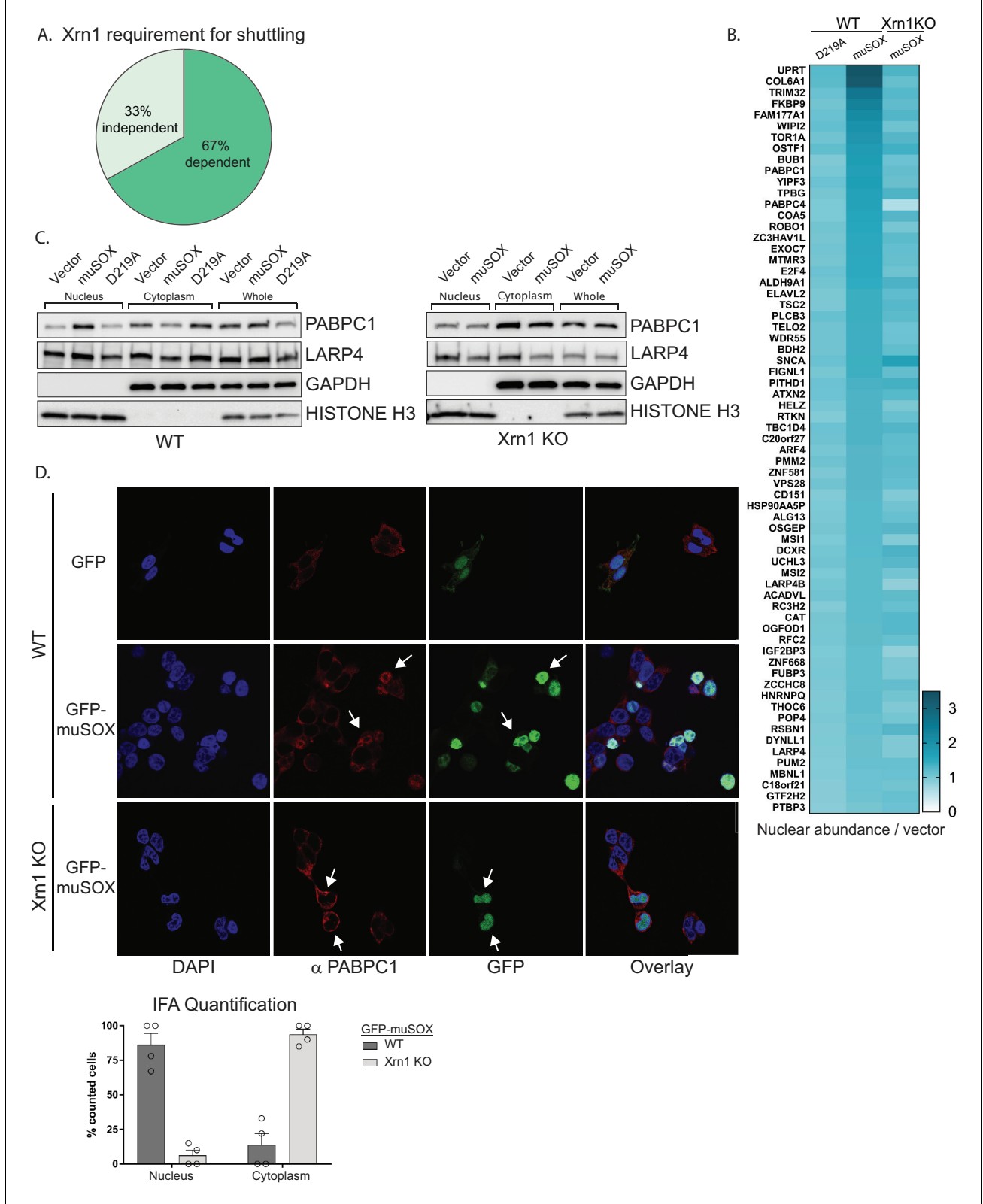

**Figure 2.** Nuclear translocation of RNA binding proteins is dependent on mRNA degradation by Xrn1. (A) Pie chart showing the percent of shuttling proteins that fail to translocate in Xrn1 KO cells. (B) Heat map depicting the average nuclear abundance in WT or Xrn1 KO HEK293T cells of the 67 significantly shifted proteins in samples expressing muSOX or D219A, relative to the empty vector control. (C) Western blots of nuclear, cytoplasmic, and whole cell fractions of WT (left panel) or Xrn1 KO (right panel) HEK293T cells transfected with the indicated plasmid. GAPDH and histone H3 serve

*Figure 2 continued on next page*

*Figure 2 continued*

as fractionation and loading controls. (D) Confocal microscopy and quantification of WT or Xrn1 KO HEK293T cells transfected with GFP or GFP-muSOX, showing signals for DAPI stained nuclei (blue), PABPC (red), GFP (green), and the merged images (overlay). Arrow heads point to representative GFP-muSOX expressing cells. The number of cells displaying either cytoplasmic or nuclear PABPC localization by immunofluorescence (IFA) was quantified for WT or Xrn1 KO cells expressing GFP-muSOX. A total of 75 GFP-muSOX WT cells and 80 GFP-muSOX Xrn1 KO cells were counted. The graph displays individual biological replicates as dots, with the mean and SEM.

DOI: https://doi.org/10.7554/eLife.37663.005

The following figure supplement is available for figure 2:

**Figure supplement 1.** Nuclear translocation of RNA binding proteins is dependent on mRNA degradation by Xrn1.
DOI: https://doi.org/10.7554/eLife.37663.006

endogenous PABPC1 and LARP4 localization by confocal microscopy and western blot analysis of fractionated cells confirmed that both proteins translocated from the cytoplasm to the nucleus upon muSOX expression in WT but not Xrn1 knockout cells (*Figure 2C–D*, *Figure 2—figure supplement 1D*).

## Xrn1 knockout leads to subcellular redistribution of proteins functionally associated with RNA

Given that increased Xrn1 activity caused nuclear translocation of mRNA-associated RBPs, we hypothesized that RBPs linked to Xrn1 function might also exhibit altered subcellular distribution in cells lacking Xrn1. We first looked broadly for proteins with reproducibly altered abundance in the nucleus or the cytoplasm of Xrn1 knockout cells relative to the vector control cells. There were 149 and 158 proteins differentially expressed in the absence of Xrn1 in the nucleus or cytoplasm, respectively (adjusted P value< 0.05) (*Figure 3A*, Table S4 in *Supplementary file 1*). Both the oligosaccharyltransferase (OST) complex and RBPs were significantly overrepresented among the set of differentially expressed proteins in each compartment (*Figure 3A*, *Figure 3—figure supplement 1A*). The significance of the OST enrichment is currently unknown, although the OST complex has been shown to be critical for infection with flaviviruses, which depend on Xrn1 for the production of a subgenomic viral noncoding RNA (*Chapman et al., 2014*; *Moon et al., 2012*). However, the RBP enrichment is in line with Xrn1 function, and it is notable that among the proteins significantly enriched in the nucleus of Xrn1 KO cells were factors that encompass the first steps of 5′−3′ mRNA decay. These included all members of the decapping complex (DCP1A, DCP1B and DCP2), factors that promote decapping complex formation (EDC3 and EDC4), and a protein that connects the decapping complex to the deadenylation machinery (PATL1) (*Figure 3B*).

We next examined whether the absence of Xrn1 also impacted the relative abundance of its known interaction partners (as listed in the BioGRID database) in the two compartments (*Figure 3C*). This did not appear to be the case in the TMT data, as the majority of known Xrn1 protein partners were expressed at normal levels in the absence of Xrn1. However, there was a significant increase in the cytoplasmic levels of UPF1, a mediator of nonsense mediated mRNA decay (NMD). Similarly, DNASE2, a nuclease which contributes to the degradation of DNA in dying cells, had increased cytoplasmic abundance. Secernin-2 (SCRN2), a protein involved in exocytosis, translocated to the nucleus in the absence of Xrn1. Conversely, PABPC4 levels were decreased in both compartments, and two centrosomal proteins CEP152 and CEP128 were reduced in the nucleus.

Finally, we considered the possibility that upon loss of Xrn1, cells might upregulate other components of the mRNA decay machinery. Perhaps surprisingly, out of all 5994 detected proteins, only three were significantly upregulated in both the nucleus and the cytoplasm of Xrn1 knockout cells: GW182, Galectin-3, and BAG1. Among these, GW182 stands out because it is a member of the miRNA-induced silencing complex (miRISC) involved in recruitment of deadenylases to initiate degradation of target mRNAs (*Figure 3D*). This increase in GW182 abundance, along with the changes to DCP2, DDX6, and PABPC4, were independently validated by western blot analysis (*Figure 3E*). To determine whether the increases in the whole cell protein abundance of GW182 and in the nuclear protein abundance in DDX6 and PATL1 occurred at the mRNA level or were a result of translational regulation, we measured steady-state mRNA expression for each of these factors by RT-qPCR. In each case, the mRNA abundance was increased in Xrn1 knockout cells compared to WT cells (*Figure 3—figure supplement 1B*). Importantly, the increases appeared specific to these

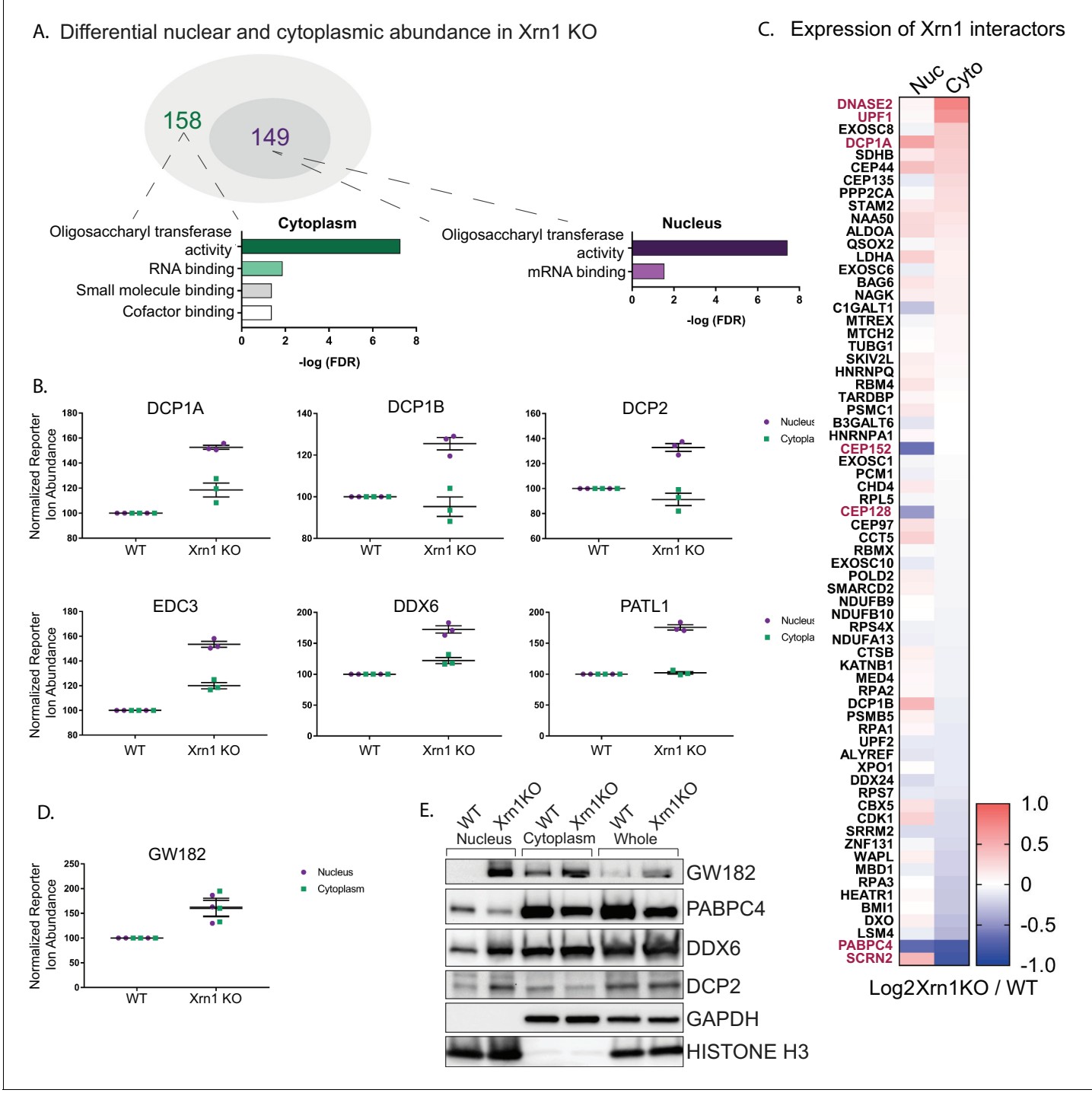

**Figure 3.** Xrn1 knockout leads to subcellular redistribution of proteins functionally associated with RNA. (**A**) The number of proteins that are differentially expressed in Xrn1 knockout (KO) cells from the nucleus (149) and the cytoplasm (158). Gene ontology molecular function overrepresentation analysis by Pantherdb is shown for each compartment, graphed according to their false discovery rate (FDR). (**B**) Graphs showing the nuclear and cytoplasmic distribution of decapping-related proteins from the TMT-LC/MS-MS data. Graphs display the mean with SEM of 3 biological replicates. (**C**) Heatmap depicting the Log$_2$ abundance ratio in Xrn1 KO HEK293T cells compared to WT HEK293T cells of proteins identified as Xrn1 interactors using the BioGRID database. Proteins with a significant difference in abundance between WT and Xrn1 KO are listed in red. (**D**) Graph of nuclear and cytoplasmic distribution of GW182 from the TMT-LC/MS-MS data. Graph displays the mean with SEM of 3 biological replicates. (**E**) Western blot of nuclear, cytoplasmic, and whole cell fractions of WT and Xrn1 KO HEK293T cells. GAPDH and histone H3 serve as fractionation and loading controls.

DOI: https://doi.org/10.7554/eLife.37663.007

*Figure 3 continued on next page*

*Figure 3 continued*

The following figure supplement is available for figure 3:

**Figure supplement 1.** Xrn1 knockout leads to subcellular redistribution of proteins functionally associated with RNA.
DOI: https://doi.org/10.7554/eLife.37663.008

transcripts and not due to generalized mRNA abundance changes in the absence of Xrn1, as there was no significant difference in *gapdh* or *actB* mRNA levels (*Figure 3—figure supplement 1B*).

Collectively, these data suggest that there are not broad increases in cellular proteins in response to inhibition of 5′−3′ mRNA decay. However, there appear to be selective increases in the whole cell or compartment-specific abundance of select factors associated with mRNA decay, which likely arises from increases in their mRNA levels in Xrn1 knockout cells.

## LARP4 shuttles to the nucleus in a PABPC-dependent manner

Protein relocalization in response to altered cytoplasmic mRNA decay could occur as a consequence of direct interactions with the nuclear transport machinery that are antagonized by mRNA, as has been documented for the PABPC nuclear localization signal (NLS) (*Kumar et al., 2011*). Alternatively, translocation could occur indirectly via interactions with other proteins that contain nuclear transport signals. To test for this latter possibility, we first plotted the network of known interactions among the list of proteins that relocalized in cells undergoing accelerated mRNA decay using the STRING database (*Figure 4A*). There were significantly more interactions among this set of proteins than would be predicted for a random group of proteins of similar size (p=0.0496), with many of the interactions involving PABPC. This enrichment suggests that these proteins are biologically related, confirming what was seen in the GO term analysis. We examined the relocalization mechanism for one of the PABPC interacting proteins, LARP4 (*Yang et al., 2011*). We reasoned that if LARP4 relocalization involved direct interactions with the nuclear import machinery, then it should relocalize in muSOX-expressing cells in a PABPC independent manner. Conversely, if it was 'escorted' into the nucleus via its interaction with PABPC, then its relocalization should be blocked by PABPC depletion. Depletion of PABPC1 has been shown to lead to compensatory induction of PABPC4, which can function in a redundant manner (*Kumar and Glaunsinger, 2010*). Therefore, we co-depleted both PABPC1 and PABPC4 using siRNAs. Upon co-depletion of the PABPC proteins, LARP4 no longer accumulated in the nucleus of muSOX-expressing cells (*Figure 4B*). In contrast, siRNA-mediated depletion of LARP4 had no effect on PABPC1 shuttling in these cells (*Figure 4C*). These results support a model in which LARP4 is brought into the nucleus in cells undergoing accelerated mRNA decay through its interaction with PABPC.

## PABPC depletion abrogates the muSOX-driven decrease in RNAPII promoter occupancy

Given the nuclear enrichment of many poly(A) and poly(U) associated proteins, we considered these factors to be strong candidates for involvement in the signaling pathway linking accelerated mRNA decay to RNAPII transcriptional repression. To determine if they were required for the mRNA decay-transcription feedback loop, we tested whether depletion of several of these factors individually altered RNAPII occupancy using chromatin immunoprecipitation assays (ChIP). To test the role of PABPC we co-depleted both PABPC1 and PABPC4 using siRNAs, then monitored RNAPII occupancy at two cellular promoters (*gapdh, rplp0*) previously shown to be responsive to mRNA decay-induced transcriptional repression (*Abernathy et al., 2015*). In cells depleted of PABPC1 and PABPC4, there was no longer a reduction in RNAPII occupancy at the *gapdh* and *rplp0* promoters in muSOX expressing cells relative to vector control cells (*Figure 5A*). In contrast, RNAPII promoter occupancy remained repressed in muSOX expressing cells upon depletion of LARP4 (*Figure 5B*). In addition to poly(A) tail associated proteins, we tested the effects of depleting three additional factors that translocated to the nucleus in an mRNA-decay dependent manner: the poly(U) binding protein MSI1, the CHD3 transcriptional regulator, and one of the top scoring hits from the MS data, TRIM32 (*Figure 5—figure supplement 1A–C*). RNAPII occupancy remained reduced in muSOX-expressing cells relative to vector control cells upon depletion of MSI1, CHD3, and TRIM32 (*Figure 5—figure supplement 1A–C*).

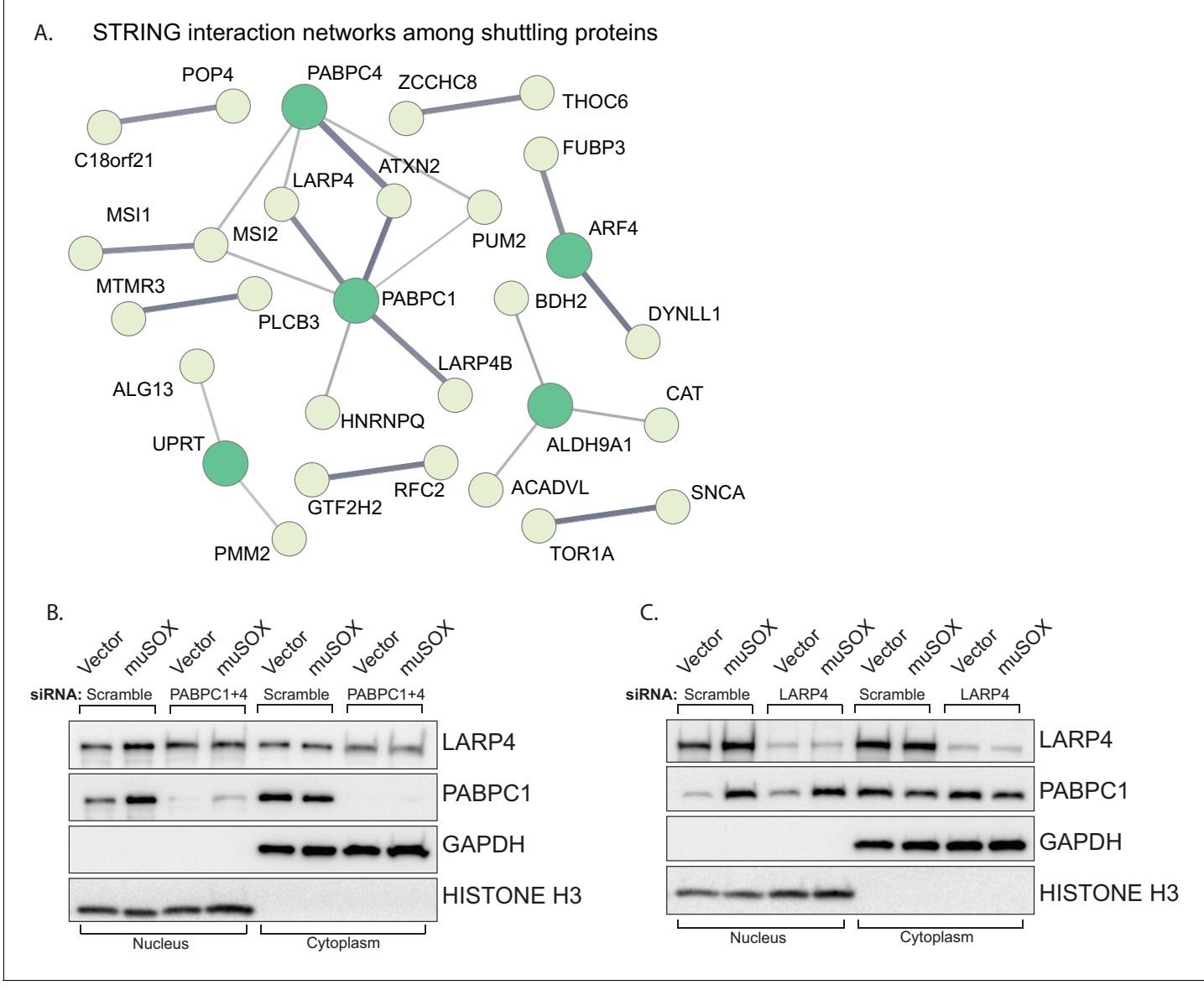

**Figure 4.** LARP4 translocates to the nucleus in a PABPC-dependent manner. (**A**) STRING network of reported protein-protein interactions between the 67 proteins that shuttle in muSOX-expressing cells. Medium and high confidence interactions are shown with thin and thick connector lines, respectively. (**B, C**) Western blots of nuclear and cytoplasmic fractions of vector- or muSOX-transfected HEK293T cells treated with the indicated siRNA. GAPDH and histone H3 serve as fractionation and loading controls.

DOI: https://doi.org/10.7554/eLife.37663.009

It should be noted that when we measured the effect of depleting the above factors on RNAPII occupancy in the absence of muSOX, we unexpectedly observed that their knockdown alone reduced the RNAPII ChIP signal (*Figure 5—figure supplement 2A–E*). However, unlike the case for PABPC, RNAPII levels were further reduced in muSOX expressing cells after depletion of LARP4, MSI1, CHD3, and TRIM32 (*Figure 5B*, *Figure 5—figure supplement 1A–C*). We hypothesize that knockdown of these factors may lead to broad impacts on cellular function, in ways that directly or indirectly influence transcription. Therefore, although PABPC appeared to be selectively involved in suppressing RNAPII occupancy during enhanced mRNA decay, we sought an alternative strategy to evaluate its connection to this process.

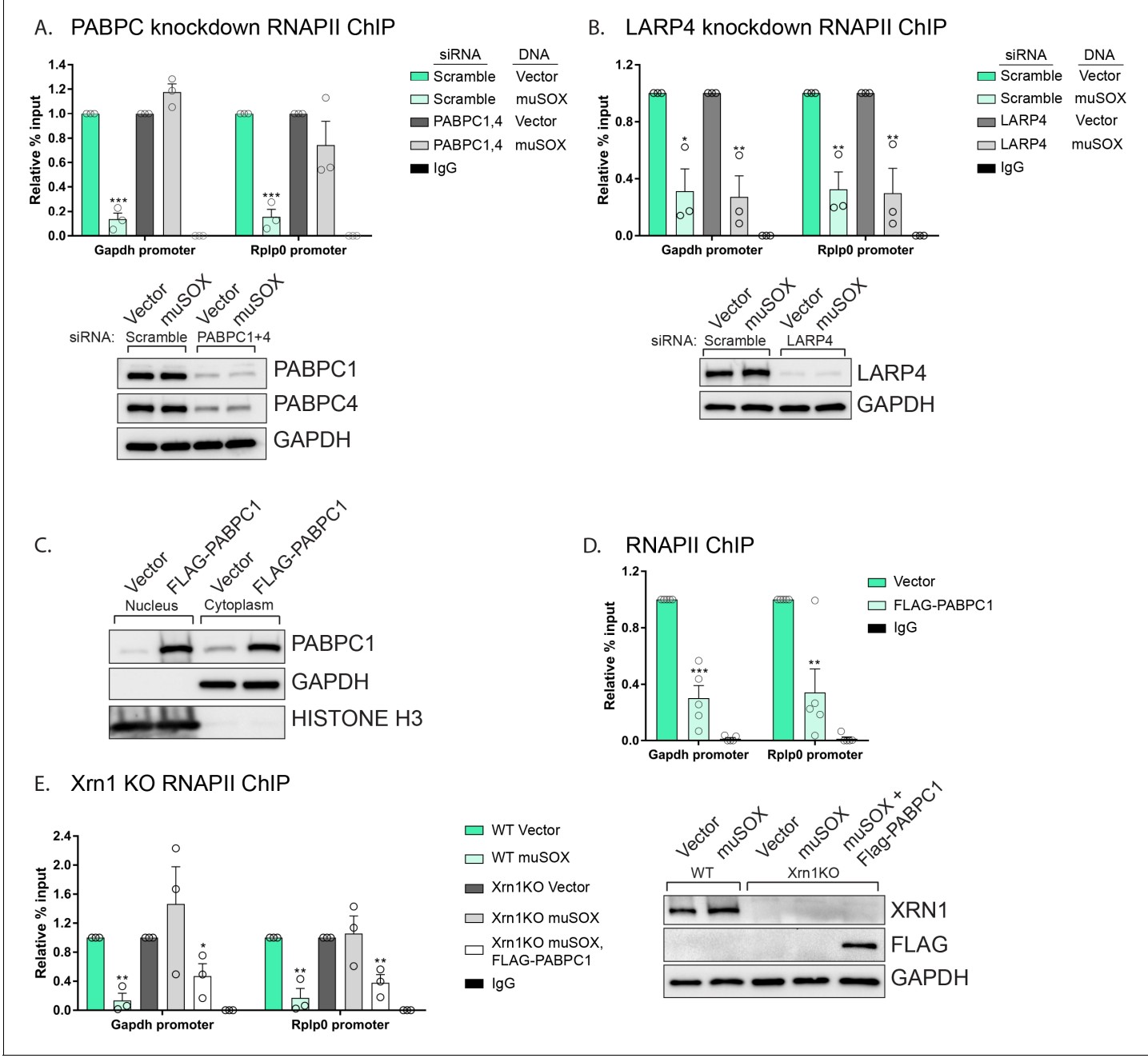

**Figure 5.** PABPC depletion prevents muSOX-induced repression of RNAPII recruitment. (**A, B**) HEK293T cells treated with siRNAs targeting PABPC1 and 4 (**A**), LARP4 (**B**), or non-targeting scramble siRNAs were subsequently transfected with either empty vector or muSOX, then subjected to chromatin immunoprecipitation (ChIP) using antibodies to RNAPII or IgG. Western blots showing protein levels of PABPC1, PABPC4, and Larp4 after siRNA depletion are shown in the lower panels, along with a GAPDH loading control. (**C**) Western blots of nuclear and cytoplasmic fractions of HEK293T cells transfected with an empty vector or a plasmid containing FLAG-PABPC1. GAPDH and histone H3 serve as fractionation and loading controls. (**D**) HEK293T cells transfected with either empty vector or FLAG-PABPC1 were subjected to ChIP using antibodies to RNAPII or IgG. (**E**) WT or Xrn1 KO HEK293T cells transfected with either empty vector or FLAG-PABPC1 alone or together with muSOX were subjected to ChIP using antibodies to RNAPII or IgG. Purified chromatin in each of the above experiments was quantified by qPCR. Western blots showing the levels of Xrn1 in WT or Xrn1KO HEK293Ts are shown, along with a GAPDH loading control. All graphs display individual biological replicates as dots, with the mean and SEM. Statistical significance was determined using Student's t test *p<0.05 **p<0.005 ***p<0.0005.
DOI: https://doi.org/10.7554/eLife.37663.010

The following figure supplements are available for figure 5:

**Figure supplement 1.** PABPC is required for the connection between cytoplasmic mRNA decay and RNAPII promoter occupancy.

*Figure 5 continued on next page*

*Figure 5 continued*

DOI: https://doi.org/10.7554/eLife.37663.011

**Figure supplement 2.** Effects of depleting PABPC, LARP4, CHD3, MSI1 and TRIM32 on RNAPII promoter occupancy.

DOI: https://doi.org/10.7554/eLife.37663.012

## Nuclear accumulation of PABPC1 is sufficient to inhibit RNAPII recruitment to promoters

Endogenous PABPC is subject to translational autoregulation, and our previous data suggested that the abundance of PABPC in uninfected cells is fine-tuned to match poly(A) tail availability (*Kumar and Glaunsinger, 2010*; *Kumar et al., 2011*). In this regard, even modest over-expression of PABPC1 leads to nuclear accumulation of the 'excess' (presumably non-poly(A) bound) protein in cells lacking muSOX (*Figure 5C*). This feature enabled us to test whether nuclear accumulation of PABPC1 was sufficient to cause a reduction in RNAPII promoter recruitment in the absence of muSOX-induced mRNA decay. Indeed, FLAG-PABPC1 transfected cells displayed a significant decrease in RNAPII occupancy at the *gapdh and rplp0* promoters (*Figure 5D*). These observations suggested that the failure of muSOX to trigger transcriptional repression in Xrn1 knockout cells might be overcome by driving PABPC into the nucleus via overexpression. In agreement with this prediction, muSOX-induced transcriptional repression was restored in Xrn1 knockout cells upon transfection of FLAG-PABPC1, confirming that nuclear translocation of this RBP plays a central role in connecting cytoplasmic mRNA decay to RNAPII promoter recruitment (*Figure 5E*).

## Nuclear translocation of PABPC selectively impacts early stages of transcription

To more precisely define the stage(s) of transcription impacted by mRNA decay-induced translocation of PABPC, we began by measuring RNAPII occupancy at both the promoter and the gene body (exon) of the genes *gapdh*, *actB*, and *tlcd1*. In each of the experiments below, we evaluated cells transfected with empty vector control, muSOX (to activate cytoplasmic mRNA decay), or FLAG-PABPC1 (to selectively increase nuclear PABPC levels in the absence of widespread mRNA decay). Cells expressing muSOX or FLAG-PABPC1 exhibited parallel phenotypes, in which RNAPII occupancy was reduced at promoters as well as within the gene body compared to control cells (*Figure 6A*). Western blotting confirmed that the reduced ChIP signals were not due to a decrease in the overall levels of RNAPII in these cells (*Figure 6—figure supplement 1*).

The C-terminal domain (CTD) of the RNAPII Rpb1 subunit has unique phosphorylation patterns associated with each phase of transcription; it initially binds DNA in an unphosphorylated state, but undergoes progressive serine 5-phosphorylation (Ser5P) during initiation, then serine 2-phosphorylation (Ser2P) during elongation (*Heidemann et al., 2013*). To determine whether mRNA decay-induced PABPC1 translocation impacted RNAPII initiation or elongation in addition to promoter recruitment, we measured the ratio of total RNAPII to either Ser5P or Ser2P RNAPII (*Figure 6B and C*). In both muSOX and FLAG-PABPC expressing cells, these ratios were unchanged relative to control cells, suggesting that the primary defect is in promoter recruitment, and that there are not independent impacts on downstream events. These data are consistent with previous observations in MHV68-infected cells (*Abernathy et al., 2015*).

RNAPII promoter recruitment occurs during assembly of the transcription preinitiation complex (PIC), a multi-step event involving numerous general transcription factors and transcription associated factors (*Roeder, 1996*). The initial promoter-defining event in PIC assembly that occurs prior to RNAPII recruitment is binding of TATA-binding protein (TBP) as part of the transcription factor TFIID complex, whose recruitment is essential for initiating transcription (*Darzacq et al., 2007*; *Louder et al., 2016*). Notably, TBP ChIP revealed that its occupancy at the *gapdh*, *actB*, *tlcd1*, and *rplp0* promoters was significantly reduced in cells expressing either muSOX or FLAG-PABPC1 compared to control cells (*Figure 6D*). Similar to RNAPII, western blotting confirmed that this reduction in promoter binding was not due to altered expression of TBP in these cells (*Figure 6—figure supplement 1*). Given that TBP is a transcription factor required by cellular polymerases other than just RNAPII, we considered the possibility that all TBP-dependent transcription might be impaired as a consequence of cytoplasmic mRNA decay. This was not the case however, as 7SK and U6 promoter

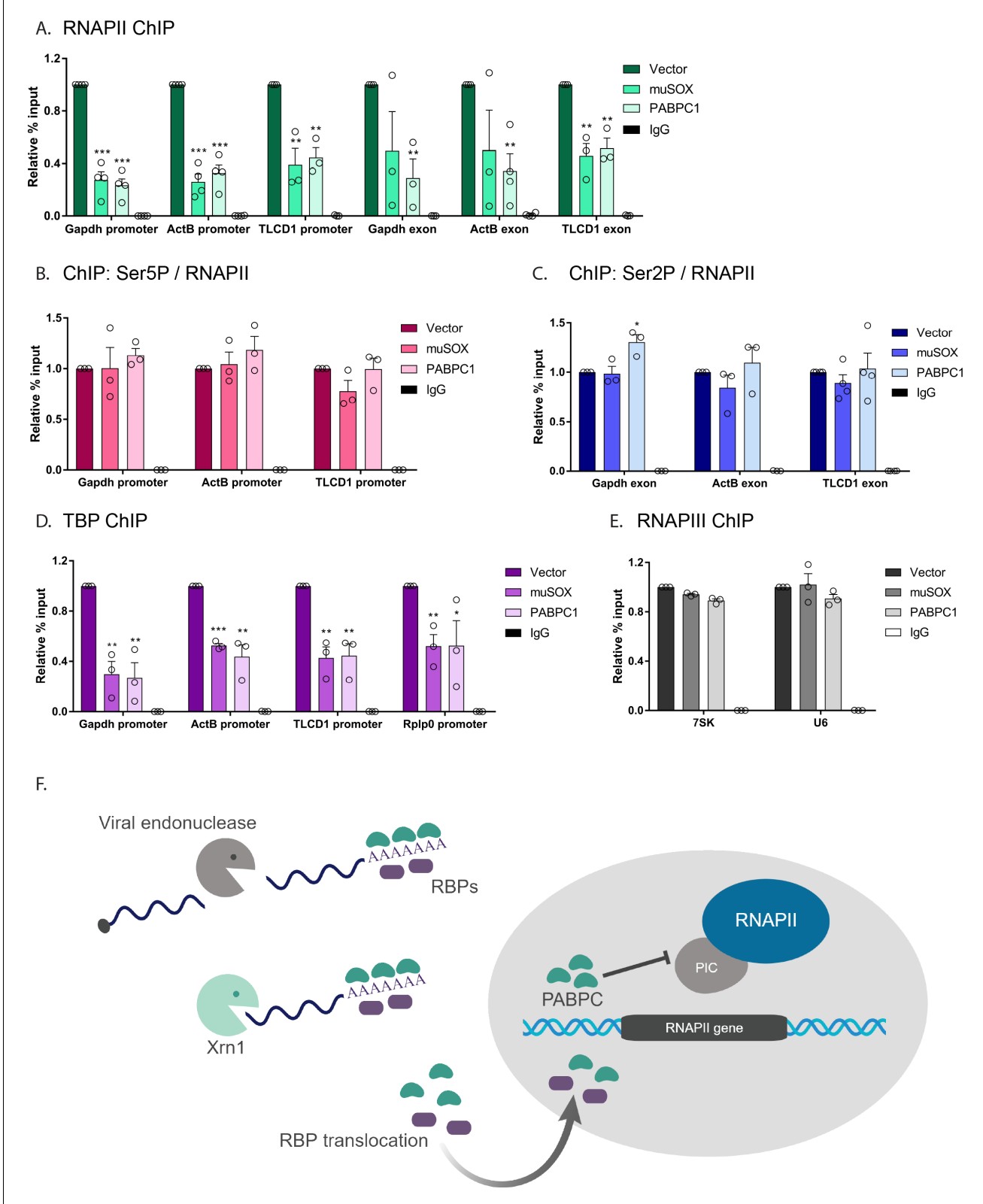

**Figure 6.** Nuclear translocation of PABPC selectively impacts early stages of transcription. (A) HEK293T cells transfected with empty vector, muSOX, or FLAG-PABPC1 were subjected to ChIP using antibodies to RNAPII or IgG at the indicated gene matched promoters and exons. (B) ChIP using antibodies to serine 5-phosphorylated (Ser5P) RNAPII or IgG at gene promoters. The level of Ser5P RNAPII was determined by dividing the Ser5P values over the total RNAPII values within the same region of the gene in HEK293T cells transfected with empty vector, muSOX, or FLAG-PABPC1. (C)

*Figure 6 continued on next page*

*Figure 6 continued*

ChIP was performed as described in (B), but using antibodies to serine 2-phosphorylated (Ser2P) RNAPII or IgG at gene exons. The level of Ser2P RNAPII was determined by dividing the Ser2P values over the total RNAPII values within the same region of the gene. (D) ChIP was performed as described in (B), but using antibodies to TATA-binding protein (TBP) or IgG at gene promoters. (E) ChIP was performed as described in (B), but using antibodies to the POLR3A subunit of RNAPIII or IgG. In each experiment, chromatin was quantified by qPCR and all graphs display individual biological replicates as dots, with the mean and SEM. Statistical significance was determined using Student's t test *p<0.05 **p<0.005 ***p<0.0005. (F) Model summarizing the impact of SOX and Xrn1-driven mRNA degradation on RBP trafficking and RNAPII transcription. See text for details.
DOI: https://doi.org/10.7554/eLife.37663.013

The following figure supplement is available for figure 6:

**Figure supplement 1.** RNAPII Rpb1 and TBP protein levels are unchanged in cells expressing muSOX or FLAG-PABPC1.
DOI: https://doi.org/10.7554/eLife.37663.014

occupancy by RNA polymerase III (RNAPIII), which also requires TBP, was unaltered in cells expressing muSOX or FLAG-PABPC1 compared to control cells (*Figure 6E*). We therefore conclude that mRNA decay-driven nuclear accumulation of PABPC1 reduces PIC assembly selectively at the promoters of RNAPII transcribed genes.

## Discussion

Cellular mRNA abundance can be dramatically altered in response to a variety of pathogenic and nonpathogenic stresses including both viral and bacterial infections, and early apoptosis (*Abernathy and Glaunsinger, 2015*; *Barry et al., 2017*; *Thomas et al., 2015*). In many of these cases, accelerated cytoplasmic mRNA decay initiates a widespread reduction in transcript levels, often through the engagement of the major mammalian 5′−3′ exonuclease Xrn1 (*Covarrubias et al., 2011*; *Gaglia et al., 2012*). In addition to altering the translational landscape, depletion of cytoplasmic mRNA elicits changes in upstream components of the mammalian gene expression pathway, including RNAPII transcription, largely by unknown mechanisms (*Abernathy et al., 2015*). Here, we tested the hypothesis that cellular RNA binding proteins may shift their subcellular localization in response to altered mRNA decay, thus conveying mRNA abundance information between the cytoplasm and the nucleus (*Figure 6F*). Quantitative proteomics was previously reported to allow the discovery of viral infection-induced protein translocations (*Jean Beltran et al., 2017*). Indeed, our unbiased TMT-based proteomics approach revealed that among the total cellular protein pool, an RBP-enriched protein subset concentrates in the nucleus specifically in response to increased mRNA decay in an Xrn1 dependent manner. RBPs have critical roles in all stages of gene expression (*Müller-McNicoll and Neugebauer, 2013*), and our data further emphasize their multifunctional capacity.

We also found that RBPs are enriched in the set of proteins with altered nuclear or cytoplasmic localization in Xrn1 knockout cells. Interestingly, factors involved in decapping, the event that directly precedes Xrn1 attack during basal mRNA decay, were selectively increased in the nuclei of cells lacking Xrn1. Furthermore, we detected increased levels of the NMD factor UPF1 in the cytoplasm and overall elevated levels of GW182 in these cells. One speculative possibility is that these changes occur in response to cellular 'reprogramming' of the mRNA decay network, for example shifting emphasis towards 3′ end targeting mechanisms to compensate for the absence of the primary 5′ end decay mechanism. This scenario might explain the increase in GW182 levels, as it recruits the cellular deadenylase complexes PAN2-PAN3 and CCR4-NOT to mRNA targets to promote mRNA decay by Xrn1 (*Braun et al., 2011*; *Chekulaeva et al., 2011*; *Fabian et al., 2011*; *Huntzinger and Izaurralde, 2011*). In its absence, the increased GW182 levels may accelerate deadenylation to instead promote 3′−5′ decay. Alternatively, the RBP nuclear and/or cytoplasmic enrichment in Xrn1 knockout cells may reflect changes that occur when the cytoplasmic mRNA decay rate is reduced. The fact that we did not observe significant redistribution of the majority of Xrn1 interacting proteins argues against a model in which physical association with Xrn1 helps control decay factor protein localization.

Aside from RBPs, there was a clear overrepresentation of the OST complex, which catalyzes co-translational N-glycosylation, among the set of differentially expressed proteins in Xrn1 knockout cells. Although OST does not have established links to RNA decay or Xrn1, it has been shown to be

a critical component of the replication cycle of flaviviruses such as Dengue and Zika (*Marceau et al., 2016*; *Puschnik et al., 2017*). Furthermore, these arthropod-borne flaviviruses inhibit Xrn1 activity through a subgenomic viral noncoding RNA that contains an Xrn1 blocking sequence (*Chapman et al., 2014*; *Moon et al., 2012*). In this context, it will be exciting to explore possible links between these two processes, especially given that small molecule OST inhibitors are now being tested for their pan-flaviviral inhibition (*Puschnik et al., 2017*).

Among the set of proteins that translocated in cells undergoing accelerated Xrn1-dependent mRNA decay, there was a striking enrichment in factors that bind the 3'end of mRNAs. This supports the hypothesis that this class of RBPs would be significantly impacted by the mRNA abundance and availability. PABPC nuclear translocation in particular has been well documented in the context of infection with viruses that drive mRNA decay (*Bablanian et al., 1991*; *Borah et al., 2012*; *Harb et al., 2008*; *Lee and Glaunsinger, 2009*; *Park et al., 2014*; *Piron et al., 1998*; *Salaun et al., 2010*), and our unbiased proteomics approach establishes it as one of the most robustly relocalized RBPs under these conditions. Several features of PABPC render it an ideal indicator of mRNA abundance. First, its association with poly(A) tails implies that depletion of mRNAs but no other type of abundant non-polyadenylated RNAs should selectively alter the level of PABPC in the RNA bound versus unbound state. Second, nuclear import of PABPC is antagonized by cytoplasmic mRNA abundance. We previously reported that PABPC harbors noncanonical NLSs within its RNA recognition motifs (RRMs); upon poly(A) binding, these elements are masked and the protein is thus retained in the cytosol (*Kumar et al., 2011*). However, release of PABPC from mRNA exposes the NLSs, enabling its interaction with importin α and its subsequent nuclear import. The observation that PABPC localization is directly influenced by mRNA abundance suggest that cells must carefully calibrate the ratio of PABPC to mRNA. Indeed, PABPC protein binds an autoregulatory A-rich sequence in the 5'UTR of its own mRNA to disrupt 40S ribosomal scanning and reduce its translation (*de Melo Neto et al., 1995*; *Wu and Bag, 1998*).

When bound to poly(A) tails in the cytoplasm, PABPC contributes to mRNA stability and facilitates protein-protein interactions for efficient translation by the ribosome (*Burgess and Gray, 2010*; *Fatscher et al., 2014*). However, when concentrated in the nucleus, PABPC functions instead to restrict gene expression. One previously established mechanism by which gene expression is inhibited involves disruption of mRNA processing, where PABPC drives hyperadenylation of nascent mRNAs (*Kumar and Glaunsinger, 2010*). In this study, we reveal that nuclear accumulation of PABPC phenotypically mimics muSOX-dependent repression of RNAPII promoter binding; it appears necessary and sufficient to repress RNAPII promoter recruitment as a consequence of accelerated mRNA decay. Both muSOX and FLAG-PABPC1 expression target early stages of PIC assembly, as TBP and RNAPII occupancy are reduced at promoters. Interestingly, the *S. cerevisiae* nuclear poly(A) binding protein Nab2 has been shown to potentiate RNAPIII activity by directly binding RNAPIII and stabilizing TFIIIB with promoter DNA (*Reuter et al., 2015*), providing a precedent for PABPs influencing transcription. However, although TBP is required for the activity of other polymerases including RNAPIII, we found that the impact of mRNA decay-induced PABPC translocation appears specific to RNAPII responsive promoters. Furthermore, while RNAPII levels are reduced at both promoters and in the gene body, the residual promoter-bound population of RNAPII does not appear to have additional defects in promoter escape or elongation, as measured by polymerase CTD phosphorylation patterns. Collectively, these observations suggest that altered PABPC trafficking primarily impacts the very earliest stages of PIC assembly. Determining which factors govern the specificity for RNAPII responsive promoters during accelerated mRNA decay and their connection to nuclear PABPC remain important challenges for the future.

Although we did not detect a role for LARP4, MSI1, CHD3, or TRIM32 in muSOX-induced transcriptional repression, these findings are complicated by the observation that their depletion alone impaired RNAPII recruitment. In our hands, this phenotype is common to the depletion of a number of different RBPs (though not all), suggesting that their absence may cause secondary effects on gene expression. This also underscores the importance of using alternative assays to evaluate their contributions, as we did for PABPC. Interestingly, some of these proteins are likely to engage in distinct gene regulatory functions in the nucleus that could also be impacted by their altered nuclear-cytoplasmic trafficking. For example, a nuclear role for MSI1 has recently been uncovered during mouse spermatogenesis, when it translocates from the cytoplasm to the nucleus (*Sutherland et al., 2015*). In the cytoplasm, MSI1 negatively regulates the translation of its target RNAs by competing

with eukaryotic initiation factor eIF4G for binding to PABPC (*Kawahara et al., 2008*). However, upon spermatocyte differentiation, MSI1 relocalizes to the nucleus through direct interaction with importin-5 (IPO5), where it concentrates at the silent XY chromatin domain. This not only releases its repression on translation, but also alters its repertoire of RNA targets in the nucleus. LARP4 also binds PABPC, but unlike MSI1, this interaction promotes mRNA poly(A) tail lengthening and stabilization in the cytoplasm (*Yang et al., 2011*). Our findings suggest that nuclear accumulation of LARP4 is also dependent on its interaction with PABPC. LARP4 protein levels are controlled post-transcriptionally via an instability determinant within its coding sequence, suggesting that akin to PABPC, its protein abundance is tightly regulated (*Mattijssen et al., 2017*). The functions of LARP4 in the nucleus, as well as other RBPs identified in this work, are currently unknown. Exploring these roles and how they become manipulated during times of cellular stress are areas ripe for future studies.

Finally, it is notable that connections between Xrn1-driven mRNA decay and RNAPII transcription have also been made in yeast, providing further evidence that these seemingly divergent stages of the gene expression cascade are intimately linked (*Haimovich et al., 2013*; *Sun et al., 2013*). However, one key difference between this pathway in yeast and mammalian cells is that in yeast it appears to operate as a compensatory mechanism to maintain optimal mRNA abundance: reduced mRNA decay results in reduced transcription, and vice versa (*Haimovich et al., 2013*; *Sun et al., 2013*). This potentially represents an evolutionary divergence in which a unicellular eukaryote 'buffers' its overall gene expression for continued maintenance of the organism. In multicellular eukaryotes like mammals, global mRNA decay (which is induced by numerous pathogens) may instead serve as a stress signal, and the ensuing response is thus geared towards shutdown of major cellular programs.

# Materials and methods

## Key resources table

| Reagent type (species) or resource | Designation | Source or reference | Identifiers | Additional information |
|---|---|---|---|---|
| Strain, strain background (murine herpesvirus 68) | MHV68 | PMID: 10888635 | NCBI_refseq ID: NC_001826.2 | Koszinowski Lab |
| Strain, strain background (murine herpesvirus 68) | R443I MHV68 | PMID: 21811408 | | Glaunsinger Lab |
| Cell line (Homo sapiens) | HEK293T | American Type Culture Collection | Cat# CRL-11268; RRID: CVCL_1926 | |
| Cell line (Mus musculus) | NIH3T3 | American Type Culture Collection | Cat# CRL-1658; RRID: CVCL_0594 | |
| Transfected construct (synthesized) | Xrn1 knockout (KO) cells | This paper | | HEK293T clone stably expressing Cas9 and Xrn1 single-guide RNA. |
| Transfected construct (synthesized) | Cas9-expressing WT cells | This paper | | HEK293T clone stably expressing Cas9 alone with no guide RNA. |
| Recombinant DNA reagent | GFP-muSOX (plasmid) | This paper | | Progenitors: pcDNA3-HA-muSOX (*Covarrubias et al., 2009*); Gateway vector peGFP-C1 |
| Recombinant DNA reagent | Thy1.1-muSOX (plasmid) | This paper | | Progenitors: GFP-muSOX |
| Recombinant DNA reagent | Thy1.1-muSOX D219A (plasmid) | This paper | | Progenitors: Thy1.1-muSOX |
| Recombinant DNA reagent | Thy1.1-GFP (plasmid) | This paper | | Progenitors: Thy1.1-muSOX |

*Continued on next page*

*Continued*

| Reagent type (species) or resource | Designation | Source or reference | Identifiers | Additional information |
|---|---|---|---|---|
| Recombinant DNA reagent | pCDEF3-FLAG-PABPC1 (plasmid) | PMID: 20823266 | | Glaunsinger Lab |
| Recombinant DNA reagent | lentiCas9-Blast (lentiviral vector) | Addgene; PMID: 25075903; PMID: 24336571 | 52962 | |
| Recombinant DNA reagent | lentiGuide-Puro (lentiviral vector) | Addgene; PMID: 25075903; PMID: 24336571 | 52963 | |
| Antibody | Mouse monoclonal anti-PABPC | Santa Cruz Biotechnologies | Clone 10 E 10; SC32318 | IFA (1:25) |
| Antibody | Rabbit polyclonal anti-LARP4 | Thermofisher | PA5-58727 | IFA (1:200); Western (1:1000) |
| Antibody | Rabbit polyclonal anti-PABPC | Cell Signaling Technology | 4992S | Western (1:1000) |
| Antibody | Rabbit polyclonal anti-PABPC4 | Bethyl | A301-466A-M | Western (1:1000) |
| Antibody | Mouse monoclonal anti-Gapdh | Abcam | Clone 6C5; ab8245 | Western (1:5000) |
| Antibody | Rabbit monoclonal anti-Histone H3 | Cell Signaling Technology | Clone D1H2; 4499S | Western (1:2000) |
| Antibody | Rabbit polyclonal anti-LYRIC | Abcam | ab124789 | Western (1:1000) |
| Antibody | Rabbit polyclonal anti-RRBP1 | Bethyl | A303-996A-T | Western (1:1000) |
| Antibody | Rabbit polyclonal anti-MSI1 | Abcam | ab52865 | Western (1:1000) |
| Antibody | Rabbit polyclonal anti-LIN28B | Abcam | ab71415 | Western (1:1000) |
| Antibody | Rabbit polyclonal anti-RPP20 | Novus Biologicals | NBP15707220 | Western (1:1000) |
| Antibody | Rabbit polyclonal anti-CHD3 | Cell Signaling Technology | 4241T | Western (1:1000) |
| Antibody | Rabbit polyclonal anti-THOC6 | Life Technologies | PA543172 | Western (1:1000) |
| Antibody | Rabbit polyclonal anti-PNN | Life Technologies | PA535053 | Western (1:1000) |
| Antibody | Rabbit polyclonal anti-EXO4 | This paper | | Western (1:1000) |
| Antibody | Rabbit polyclonal anti-NPM | Abcam | ab10530 | Western (1:1000) |
| Antibody | Rabbit polyclonal anti-TRIM32 | Abcam | ab131223 | Western (1:1000) |
| Antibody | Mouse monoclonal anti-GW182 | Abcam | Clone 4B6; ab70522 | Western (1:1000) |
| Antibody | Rabbit polyclonal anti-DDX6 | Bethyl | A300-460 | Western (1:1000) |
| Antibody | Rabbit polyclonal anti-DCP2 | Bethyl | A302-597A-M | Western (1:1000) |
| Antibody | Mouse monoclonal anti-RNAPII Rpb1 | BioLegend | 8WG16 | Western (1:2000); ChIP (10 mg) |
| Antibody | Rabbit polyclonal anti-TBP | Abcam | ab28175 | Western (1:2000); ChIP (10 mg) |

*Continued on next page*

*Continued*

| Reagent type (species) or resource | Designation | Source or reference | Identifiers | Additional information |
|---|---|---|---|---|
| Antibody | Rabbit polyclonal anti-RNAPII phospho S5 | Abcam | ab5131 | ChIP (10 mg) |
| Antibody | Rabbit polyclonal anti-RNAPII phospho S2 | Abcam | ab5095 | ChIP (10 mg) |
| Antibody | Rabbit polyclonal anti-POLR3A | Abcam | ab96328 | ChIP (10 mg) |
| Sequence-based reagent (H. sapiens) | siPABPC1 | PMID: 20823266 | | Custom siRNA; see Table S5 for sequence |
| Sequence-based reagent (H. sapiens) | siPABPC4 | PMID: 20823266 | | Custom siRNA; see Table S5 for sequence |
| Sequence-based reagent (H. sapiens) | siLARP4 | Dharmacon | M-016523-00-0020 | |
| Sequence-based reagent (H. sapiens) | siCHD3 | Dharmacon | M-023015-01-0020 | |
| Sequence-based reagent (H. sapiens) | siMSI1 | Dharmacon | M-011338-01-0010 | |
| Sequence-based reagent (H. sapiens) | siTRIM32 | Dharmacon | M-006950-01-0010 | |
| Commercial assay or kit | Protein G dynabeads | Thermofisher | 10004D | |
| Commercial assay or kit | Protein A dynabeads | Thermofisher | 10002D | |
| Commercial assay or kit | 10-plex TMT kit | Thermofisher | 90113 | |
| Chemical compound, drug | Alexa Fluor 594, goat anti-rabbit | Thermofisher | A-11072 | |
| Chemical compound, drug | Alexa Fluor 594, goat anti-mouse | Thermofisher | A-11020 | |
| Software, algorithm | Proteome Discoverer | Thermofisher | v2.2.0.388 | |

## Plasmids

Primers used for cloning are listed in Table S5 in *Supplementary file 1*. MHV68 muSOX was cloned into the Gateway entry vector pDON207 (Invitrogen), and then transferred into the Gateway-compatible peGFP-C1 destination vector to generate GFP-muSOX. Thy1.1-muSOX was generated by Infusion cloning (Clontech) of Thy1.1 (CD90.1) followed by a self-cleaving 2A peptide from foot-and-mouth disease virus in place of GFP into the Nhe1 and SacII restriction enzyme sites of GFP-muSOX. The D219A muSOX mutant was made using Quikchange site-directed mutagenesis (Agilent). Thy1.1-GFP was created with Infusion cloning to insert GFP back into the vector with the BamHI and EcoRI restriction enzyme sites to replace muSOX with GFP. pCDEF3-Flag-PABPC1 was described previously (*Kumar and Glaunsinger, 2010*). The Cas9 (lentiCas9-Blast) and sgRNA (lentiGuide-Puro) viral vectors were made as previously described (*Sanjana et al., 2014*; *Shalem et al., 2014*). The Xrn1 sgRNA was chosen using the Broad sgRNA design website (*Doench et al., 2014*).

## Cells and transfections

NIH3T3 cells and HEK293T cells, both from ATCC and obtained through the UC Berkeley Tissue Culture Facility, were maintained in DMEM (Invitrogen) supplemented with 10% fetal bovine serum. Cell lines were authenticated by STR analysis, and determined to be free of mycoplasma by PCR screening. DNA transfections were carried out in HEK293T cells at 70% confluency in 15 cm plates with 25 μg DNA using PolyJet (SignaGen) for 24 hr. For small interfering RNA (siRNA) transfections, HEK293T cells were transfected twice over 48 hr with 100 μM siRNA using Lipofectomine RNAiMAX (Thermo Fisher), whereupon the cells were transfected with the indicated DNA plasmid for an

additional 24 hr. Non-targeting scramble siRNAs, LARP4, MSI1, CHD3, and TRIM32 siRNAs were obtained from Dharmacon (Scramble: D-001206-13-50, LARP4: M-016523-00-0020, MSI1: M-011338-01-0010, CHD3: M-023015-01-0020, TRIM32: M-006950-01-0010). PABPC1 and PABPC4 siRNAs have been previously described and are listed in Table S5 in *Supplementary file 1* (*Kumar and Glaunsinger, 2010*; *Lee and Glaunsinger, 2009*).

The Xrn1 knockout clone and control Cas9-expressing cells were made by transducing HEK293T cells as previously described (*Sanjana et al., 2014*; *Shalem et al., 2014*). Briefly, lenti-Cas9-blast len- tivirus was spinfected onto a monolayer of HEK293T cells, which were then incubated with 20 µg/ml blasticidin to remove non-transduced cells. These Cas9-expressing HEK293T cells were then spin- fected with lentivirus made from lentiGuide-Puro containing the Xrn1 sgRNA sequence and selected with 1 µg/ml puromycin. The pool of Xrn1 knockout cells was then single-cell cloned in 96-well plates and individual clones were screened by western blot to determine knockout efficiency.

Pure populations of cells expressing muSOX were generated using the Miltenyi Biotec MACS cell separation system. HEK293T cells were transfected with either Thy1.1-GFP, Thy1.1-muSOX, or Thy1.1-muSOX D219A for 24 hr, whereupon cells were washed twice with PBS and cell pellets were resuspended in 95 µl auto-MACS rinsing buffer supplemented with 0.5% FBS and incubated with 3 µl anti-CD90.1 microbeads on ice for 10 – 15 min, and mixed by flicking the tube every 5 min. Cells were then magnetically separated according to the manufacturer's instructions. Thy1.1 positive cells were used in all downstream experiments unless otherwise stated.

## Viruses and infections

The MHV68 bacterial artificial chromosome (BAC), and the construction of the R443I muSOX mutant were previously described (*Adler et al., 2000*; *Richner et al., 2011*). MHV68 was produced by trans- fecting NIH3T3 cells in 6-well plates with 2.5 µg BAC DNA using Mirus TransIT-X2 (Mirus Bio) for 24 hr, whereupon the cells were split into a 10 cm dish and harvested after 5–7 days, once all the cells were green and dead. Virus was amplified in NIH 3T12 cells and titered by plaque assay. Cells were infected with MHV68 at an MOI of 5 for 24 hr.

## Immunofluorescence assays

Cells were plated on coverslips coated with 100 ug/mL poly-L-lysine and transfected at 70% conflu- ency with either GFP or GFP-muSOX for 24 hr. Transfected cells were fixed in 4% formaldehyde, per- meabilized with ice-cold methanol, and incubated with blocking buffer [1% Triton X-100, 0.5% Tween-20, 3% Bovine Serum Albumin] prior to incubation with mouse monoclonal PABPC diluted 1:25 (Santa Cruz Biotechnologies, 10E10) or rabbit polyclonal LARP4 diluted 1:200 (Thermo Fisher) in blocking buffer at 4°C overnight, followed by incubation with Alexa Fluor 594-conjugated goat anti-mouse, or anti-rabbit secondary antibody (Thermo Fisher, 1:1000) and DAPI (Pierce, 1:1000). Coverslips were mounted on slides using Vectashield hard-set mounting medium (VectorLabs) and imaged by confocal microscopy on a Zeiss LSM 710 AxioObserver microscope.

## Fractionation

HEK293T cells were fractionated using the REAP method (*Nabbi and Riabowol, 2015*). Briefly, cells were washed twice with ice-cold PBS and the cell pellet was lysed in 0.1% NP-40 PBS lysis buffer. The nuclei were then isolated by differential centrifugation at 10,000 x *g* for 10 s and the supernatant retained as the cytoplasmic fraction. For western blotting, the nuclei were sonicated in 0.1% NP-40 PBS lysis buffer.

## Cell lysis and protein digestion

WT Cas9-HEK293T cells were transfected with Thy1.1-GFP, Thy1.1-muSOX, or Thy1.1-muSOX D219A. Xrn1 knockout HEK293T cells were transfected with Thy1.1-GFP or Thy1.1-muSOX for 24 hr, followed by Thy1.1 separation. Separated cells were then fractionated as described above, and nuclear pellets were snap-frozen in liquid nitrogen. Cytoplasmic fractions were concentrated using an Amicon ultra filtration unit with a molecular weight cutoff of 3 kDa (Millipore) and exchanged into a 50 mM $NH_4HCO_3$, 2% Deoxycholate buffer and then snap frozen in liquid nitrogen. The nuclear pellets were lysed in 200 µL of 100 mM Tris-HCl, pH 8.0, 4% SDS, 1 mM EDTA preheated to 70°C. Cytoplasmic fractions were thawed and adjusted to 1% SDS with a 10% SDS solution. Complete lysis

of samples was achieved via five successive rounds of heating at 95°C for 3 min followed by sonication for 10 s in a cup horn sonicator set on 1 s pulses at medium output. Protein amounts were assessed by BCA protein assay (Pierce) and 50 µg of protein from each sample was simultaneously reduced and alkylated with 20 mM tris(2-carboxyethyl)phosphine (Pierce) and chloroacetamide respectively for 20 min at 70°C. Protein samples were then cleaned up by methanol-chloroform precipitation (*Wessel and Flügge, 1984*; Federspiel and Cristea, 2018, In press). LC-MS grade methanol, chloroform, and water (at a 4:1:3 ratio) were added to the sample with vortexing following each addition. The samples were spun at 2,000 × $g$ for 5 min at room temperature and the top phase was removed. Three volumes of cold methanol were then added and the samples were spun at 9,000 × $g$ for 2 min at 4°C. All liquid was removed and the protein pellets were washed with five volumes of cold methanol and then spun at 9,000 × $g$ for 2 min at 4°C. All liquid was removed again and the dried protein pellets were resuspended in 50 mM HEPES pH 8.5 at a 0.5 µg/µL concentration. Trypsin (Pierce) was added at a 1:50 trypsin:protein ratio and the samples were incubated at 37°C overnight.

## TMT labeling

Digested samples were concentrated by speed vac to one half the original volume prior to labeling and adjusted to 20% acetonitrile (ACN). All three biological replicates were labeled concurrently with a 10-plex TMT kit (Thermo Fisher Scientific) as in (*Sauls et al., 2018*). The TMT reagents (0.8 mg per channel) were dissolved in 42 µL of anhydrous ACN and 14 µL of this was added to each sample following the scheme in *Figure 1A* and allowed to react at RT for 1 hr. The labeling was quenched by the addition of hydroxylamine to a final 0.5% (v/v) concentration followed by incubation at RT for 15 min. Labeled peptides were pooled at equal peptide amounts thereby generating three 10-plex experiments, each of which was an individual biological replicate. An initial test mix for each replicate was analyzed, and the apparent peptide ratios were determined. Mixing ratios were adjusted using the information from the test mix to correct for sample losses and generate mixes with equal peptide amounts per channel.

## Peptide fractionation

Pooled peptides were acidified and fractionated by 2D StageTip (*Sauls et al., 2018*). Peptides were first desalted via C18 StageTips to remove unreacted TMT reagent by washing the bound peptides with 5% ACN, 0.5% formic acid (FA) and then eluting the peptides in 70% ACN, 0.5% FA. The eluted peptides were then bound to SCX StageTips and eluted in four fractions with sequential elution (100 µL) as follows: (1) 0.05 M ammonium formate/20% ACN, (2) 0.05 M ammonium acetate/20% ACN, (3) 0.05 M ammonium bicarbonate/20% ACN, and (4) 0.1% ammonium hydroxide/20% ACN. Each of these fractions were diluted 1:1 with 1% trifluoroacetic acid and further fractionated by SDB-RPS StageTips with sequential elution (50 µL) into three fractions as follows: (1) 0.2 M ammonium formate/0.5% FA/60% ACN, (2) 0.2 M ammonium acetate/0.5% FA/60% ACN, (3) 5% ammonium hydroxide/80% ACN. The resulting 12 fractions for each 10-plex experiment were dried *in vacuo* and resuspended in 5 µL of 1% FA, 1% ACN in water.

## LC-MS/MS analysis

Peptides (2 µL) were analyzed by LC-MS/MS using a Dionex Ultimate 3000 UPLC coupled online to an EASYSpray ion source and Q Exactive HF. Peptides were separated on an EASYSpray C18 column (75 µm x 50 cm) heated to 50°C using a linear gradient of 5% ACN to 42% ACN in 0.1% FA over 150 min at a flow rate of 250 nL/min and ionized at 1.7kv. MS/MS analysis was performed as follows: an MS1 scan was performed from 400 to 1800 m/z at 120,000 resolution with an automatic gain control (AGC) setting of 3e6 and a maximum injection time (MIT) of 30 ms recorded in profile. The top 18 precursors were then selected for fragmentation and MS2 scans were acquired at a resolution of 60,000 with an AGC setting of 2e5, a MIT of 105 ms, an isolation window of 0.8 m/z, a fixed first mass of 100 m/z, normalized collision energy of 34, intensity threshold of 1e5, peptide match set to preferred, and a dynamic exclusion of 45 s recorded in profile.

## Informatic analysis of TMT data

MS/MS data were analyzed by Proteome Discoverer (Thermo Fisher Scientific, v2.2.0.388). The nuclear channels (126 – 128C) and cytoplasmic channels (129 N-131) were analyzed in separate Proteome Discoverer studies to not bias the quantitation due to the expected protein expression differences between these two compartments. The Spectrum Files RC node was utilized to perform post-acquisition mass recalibration and the recalibrated spectra were passed to Sequest HT where two successive rounds of searching were employed against a Uniprot human database appended with common contaminants (2016 – 04, 22,349 sequences). Both search rounds required 5ppm accuracy on the precursor and 0.02 Da accuracy on the fragments and included static carbamidomethyl modifications to cysteine, static TMT additions to peptide N-termini and lysine residues, dynamic oxidation of methionine, dynamic deamidation of asparagine, and dynamic methionine loss and acetylation of protein n-termini. The first Sequest HT search was for fully tryptic peptides only and any unmatched spectra were sent to a second Sequest HT search, which allowed semi-tryptic peptide matches. All matched spectra were scored by Percolator and reporter ion signal-to-noise (S/N) values were extracted (The et al., 2016). The resulting peptide spectrum matches were parsimoniously assembled into a set of identified peptide and protein identifications with a false discovery rate of less than 1% for both the peptide and protein level and at least two unique peptides identified per protein. TMT reporter ion quantification was performed for unique and razor peptides with an average S/N of at least 10 and a precursor co-isolation threshold of less than 30% which did not contain a variable modification. Reporter ion values were normalized to the total detected signal in each channel and protein abundances were calculated as the sum of all normalized reporter ion values for each channel in each protein. Missing values were input using the low abundance resampling algorithm. The reporter ion values for the empty vector WT samples (channels 126 and 129N) were set as 100 and the other channels were scaled to this value. Statistically differential proteins were assessed via a background based ANOVA analysis implemented in Proteome Discoverer. Proteins and associated TMT reporter ion abundances and adjusted p-values from the ANOVA analysis were exported to Excel for further analysis. The mass spectrometry proteomics data reported in this paper have been deposited at the ProteomeXchange Consortium via the PRIDE partner repository (*Vizcaíno et al., 2014*). The PRIDE accession number is PXD009487.

## Gene Ontology analysis and informatic software used

Differential proteins (adjusted p-value$\leq$0.05) were analyzed via over representation analysis (www.pantherdb.org) for associated gene ontology enrichments (*Mi et al., 2016*). Example proteins of different classes, along with all heatmaps, were graphed in GraphPad Prism v7.

## Western blotting

Nuclear, cytoplasmic, and whole cell lysates were quantified by Bradford assay and resolved by SDS-PAGE and western blotted with antibodies against PABPC (Cell Signaling, 1:1000), PABPC4 (Bethyl, 1:1000), LARP4 (Thermo Fisher, 1:1000), Gapdh (Abcam, 1:3000), Histone H3 (Cell Signaling, 1:2000), LYRIC (Abcam, 1:1000), RRBP1 (Bethyl, 1:1000), MSI1 (Abcam, 1:1000), Lin28b (Abcam, 1:1000), CHD3 (Cell Signaling, 1:1000), RPP20 (Novus, 1:1000), THOC6 (Life Technologies, 1:1000), PNN (Life Technologies, 1:1000), EXO4 (rabbit polyclonal produced using recombinant EXO4 with an MBP tag, 1:1000), NPM (Abcam, 1:1000), GW182 (Abcam, 1:1000), DDX6 (Bethyl, 1:1000), DCP2 (Bethyl, 1:1000), TRIM32 (Abcam, 1:1000), RNAPII Rpb1 (BioLegend, 1:2000), TBP (Abcam, 1:2000).

## Chromatin immunoprecipitation (ChIP)

ChIP was performed on 15 cm plates of HEK293T cells transfected twice 4 hr apart with the indicated plasmid DNA. 24 hr after the first transfection, cells were crosslinked in 1% formaldehyde in PBS for 10 min at room temperature, quenched in 0.125 M glycine, and washed twice with ice-cold PBS. Crosslinked cell pellets were mixed with 1 ml ice-cold ChIP lysis buffer (5 mM PIPES pH 8.0, 85 mM KCl, 0.5% NP-40) and incubated on ice for 10 min, whereupon the lysate was dounce homogenized to release nuclei and spun at 1.5 x *g* for 5 min at 4°C. Nuclei were then resuspended in 500 μl of nuclei lysis buffer (50 mM Tris-HCl pH 8.0, 0.3% SDS, 10 mM EDTA) and rotated for 10 min at 4°C followed by sonication using a QSonica Ultrasonicator with a cup horn set to 75 amps for 20 min total (5 min on, 5 min off). Chromatin was spun at 16,000 x *g* for 10 min at 4°C and the pellet was

discarded. 100 µl of chromatin was diluted 1:5 in ChIP dilution buffer (16.7 mM Tris-HCl pH 8.0, 1.1% Triton X-100, 1.2 mM EDTA, 167 mM NaCl) and incubated with 10 µg mouse monoclonal anti-RNAPII (BioLegend, 8WG16), rabbit IgG (Fisher Scientific), rabbit polyclonal anti-RNAPII phospho S5 (Abcam ab5131), rabbit polyclonal anti-RNAPII phospho S2 (Abcam ab5095), rabbit polyclonal anti-TBP (Abcam ab28175), or rabbit polyclonal anti-POLR3A (Abcam ab96328) overnight, whereupon samples were rotated with 20 µl protein G dynabeads (with mouse antibodies), or 20 µl mixed protein G and A dynabeads (with rabbit antibodies) (Thermofisher) for 2 hr at 4°C. Beads were washed with low salt immune complex (20 mM Tris pH 8.0, 1% Triton-x-100, 2 mM EDTA, 150 mM NaCl, 0.1% SDS), high salt immune complex (20 mM Tris pH 8.0, 1% Triton-x-100, 2 mM EDTA, 500 mM NaCl, 0.1% SDS), lithium chloride immune complex (10 mM Tris pH 8.0, 0.25 M LiCl, 1% NP-40, 1% Deoxycholic acid, 1 mM EDTA), and Tris-EDTA for 5 min each at 4°C with rotation. DNA was eluted from the beads using 100 µl of elution buffer (150 mM NaCl, 50 µg/ml proteinase K) and incubated at 50°C for 2 hr, then 65°C overnight. DNA was purified using a Zymo Oligo Clean and Concentrator kit. Purified DNA was quantified by qPCR using iTaq Universal SYBR Mastermix (BioRad) with the indicated primers (Table S5 in *Supplementary file 1*). Each sample was normalized to its own input.

## Replicates

In this study, individual biological replicates are experiments performed separately on biologically distinct samples representing identical conditions and/or time points. For cell culture-based assays, this means that the cells are maintained in different flasks. Technical replicates are experiments performed on the same biological sample multiple times. See Figure Legends for the number of experimental replicates performed for each experiment. No outliers were encountered in this study. Criteria for the inclusion of data was based on the performance of positive and negative controls within each experiment.

## Acknowledgements

This work was funded by NIH grant CA136367 to BG, who is also an investigator of the Howard Hughes Medical Institute, and by NIH GM114141 to IMC. We thank all members of the Glaunsinger and Coscoy labs for helpful comments and suggestions.

## Additional information

### Funding

| Funder | Grant reference number | Author |
| --- | --- | --- |
| National Institutes of Health | CA136367 | Britt Glaunsinger |
| Howard Hughes Medical Institute | | Britt Glaunsinger |
| National Institutes of Health | GM114141 | Ileana M Cristea |

The funders had no role in study design, data collection and interpretation, or the decision to submit the work for publication.

### Author contributions

Sarah Gilbertson, Conceptualization, Data curation, Formal analysis, Validation, Investigation, Visualization, Methodology, Writing—original draft, Writing—review and editing; Joel D Federspiel, Data curation, Formal analysis, Investigation, Methodology, Writing—review and editing; Ella Hartenian, Validation, Writing—review and editing; Ileana M Cristea, Conceptualization, Supervision, Funding acquisition, Methodology, Project administration, Writing—review and editing; Britt Glaunsinger, Conceptualization, Resources, Supervision, Funding acquisition, Project administration, Writing—review and editing

Author ORCIDs
Sarah Gilbertson (iD) http://orcid.org/0000-0003-0125-593X
Britt Glaunsinger (iD) http://orcid.org/0000-0003-0479-9377

Decision letter and Author response
Decision letter https://doi.org/10.7554/eLife.37663.020
Author response https://doi.org/10.7554/eLife.37663.021

## Additional files

### Supplementary files

• Supplementary file 1. contains the following:
DOI: https://doi.org/10.7554/eLife.37663.015
• Transparent reporting
DOI: https://doi.org/10.7554/eLife.37663.016

### Data availability

Mass spectrometry proteomics data reported in this paper have been deposited at the ProteomeXchange Consortium via the PRIDE partner repository under accession number PXD009487.

The following dataset was generated:

| Author(s) | Year | Dataset title | Dataset URL | Database, license, and accessibility information |
|---|---|---|---|---|
| Gilbertson S, Federspiel JD, Hartenian E, Cristea IM, Glaunsinger B | 2018 | Changes in mRNA abundance drive shuttling of RNA binding proteins, linking cytoplasmic RNA degradation to transcription | http://proteomecentral.proteomexchange.org/cgi/GetDataset?ID=PXD009487 | Publicly available at ProteomeXchange (accession no. PXD009487) |

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
