## [Decision Letter]

Thank you for submitting your article "Changes in mRNA abundance drive shuttling of RNA binding proteins, linking cytoplasmic RNA degradation to transcription" for consideration by *eLife*. Your article has been reviewed by three peer reviewers, one of whom is a member of our Board of Reviewing Editors, and the evaluation has been overseen by a Reviewing Editor and James Manley as the Senior Editor. The reviewers have opted to remain anonymous.

The reviewers have discussed the reviews with one another and the Reviewing Editor has drafted this decision to help you prepare a revised submission.

The present study investigates the underlying RBP-associated mechanism(s) that influence changes in cellular gene expression by altering RNA decay via the inclusion of the SOX exonuclease as well as Xrn1 knockout. The authors take an attractive MS approach and identify a set of RBPs whose subcellular localization changes in response to muSOX activity. They eventually focus on PABPC1 and provide evidence to suggest that the nuclear relocalization of this protein may be, at least partly, responsible for transcriptional changes that occur in response to SOX activity and linking cytoplasmic decay rates to nuclear transcription.

Taken together, the findings are convincing and presented clearly. The weak point is that the authors do not explain mechanistically how PABPC1 influences RNAPII activity. Indeed, they argue that the effect might be indirect. Even though a detailed molecular mechanism might be beyond the scope of the study, some description of the nature of the transcriptional shut-down should be included.

Essential revisions:

Some description of the nature of the transcriptional shut-down should be provided. This could for example be more precisely describing the generality of the RNAP down regulation. For instance, a genome-wide view of RNAPII occupancy in muSOX cells as well as in PABPC1 overexpression conditions would address the robustness of the observation and how interconnected the two phenotypes are (number of common genes, types of genes, etc.). Alternatively, or in addition, the authors could examine the elongation and termination status of affected polymerase on a few model genes. ChIP analysis using additional amplicons than the promoter amplicon already used, together with ChIP of initiation and elongation factors would help to better understand the nature of the transcriptional shut-down (i.e. transcription induction, promoter-proximal stalling, etc.). We also note, that in comparing conditions, carefully controlled experimentation is warranted; for example using positive and negative probes and normalizing to both lowly and highly expressed genes that do not show transcription variation between the conditions. A calibration spike could also be used to control for a possible bias in the chip efficiency between samples.

---

## [Author Response]

Essential revisions:Some description of the nature of the transcriptional shut-down should be provided. This could for example be more precisely describing the generality of the RNAP down regulation. For instance, a genome-wide view of RNAPII occupancy in muSOX cells as well as in PABPC1 overexpression conditions would address the robustness of the observation and how interconnected the two phenotypes are (number of common genes, types of genes, etc.). Alternatively, or in addition, the authors could examine the elongation and termination status of affected polymerase on a few model genes. ChIP analysis using additional amplicons than the promoter amplicon already used, together with ChIP of initiation and elongation factors would help to better understand the nature of the transcriptional shut-down (i.e. transcription induction, promoter-proximal stalling, etc.). We also note, that in comparing conditions, carefully controlled experimentation is warranted; for example using positive and negative probes and normalizing to both lowly and highly expressed genes that do not show transcription variation between the conditions. A calibration spike could also be used to control for a possible bias in the chip efficiency between samples.

Thank you for the suggestions; we agree that a more detailed analysis of how PABPC impacts RNAPII transcription would strengthen the story. While a genome-wide analysis of PABPC’s influence is something we are planning as part of a follow-up study (and we agree the controls mentioned above are required for this type of experiment), completing this will take many more months of work and it is beyond the scope of the current manuscript. However, we have performed a number of experiments (comprising the new Figure 6) to better describe the nature of the transcriptional inhibition:

We added an analysis of RNAPII occupancy during expression of both muSOX and FLAG-PABPC1 at two additional promoters and three additional gene exons. We find that expression of FLAG-PABPC1 very closely mimics the effects of muSOX-induced repression of RNAPII occupancy at all locations tested (Figure 6A).

We examined the initiation and elongation status of RNAPII in cells expressing either muSOX or FLAG-PABPC1. This was done by ChIP with antibodies specific for the serine 5 or serine 2 phosphorylated versions of RNAPII within three different promoters and gene exons (Figure 6B-C). For both muSOX and FLAG-PABPC expressing cells, we observed that the primary defect is at the stage of RNAPII recruitment, rather than to the subsequent initiation and elongation phases of transcription.

We performed ChIP against the initiation factor TATA-binding protein (TBP) in muSOX and FLAG-PABPC expressing cells (Figure 6D). Notably, TBP promoter occupancy was also significantly reduced, revealing that the impact on transcription occurs at a very early stage of preinitiation complex assembly.

To test whether the reduced RNAPII and TBP occupancy at promoters was a consequence of decreased expression of either protein, we performed western blots of the RNAPII subunit Rpb1 and TBP in control, muSOX, and FLAG-PABPC expressing cells (Figure 6—figure supplement 1). In each sample, the levels of these proteins were equivalent.

Finally, to address whether muSOX and FLAG-PABPC specifically impact RNAPII responsive promoters, we performed ChIP for the POLR3A subunit of RNAPIII. Unlike the case for the RNAPII promoters, we found no change in RNAPIII occupancy at the 7SK or U6 promoters in the presence of either protein, confirming that the defect we observe is specific to RNAPII transcription (Figure 6E).

We hope you agree that these new data provide deeper insights into the stage of transcription that is impacted by muSOX and nuclear PABPC, as well as indicate a specificity of the effect for RNAPII responsive promoters. Importantly, for each of the experiments conducted, we observed parallel phenotypes for muSOX and FLAG-PABPC expression, bolstering the hypothesis that PABPC translocation is an important effector of the mRNA decay-transcription feedback mechanism.